# Fluorine-Containing Flow Modifier for BN/PPS Composites Enabled by Low Surface Energy

**DOI:** 10.3390/molecules27228066

**Published:** 2022-11-20

**Authors:** Bo Cao, Xiaodan Huang, Wenxiang Zhang, Peng Wu

**Affiliations:** 1School of Materials Science and Engineering, South China University of Technology, Guangzhou 510640, China; 2Key Lab Guangdong High Property & Functional Polymer Materials, and Key Laboratory of Polymer Processing Engineering, Ministry of Education, Guangzhou 510640, China

**Keywords:** BN/PPS composites, fluorine-containing flow modifier, dispersion, thermal conductivity, dielectric property

## Abstract

In this study, a fluorine-containing flow modifier (Si-DF) with low surface energy is successfully synthesized, which is applied to fabricate ideal electronic packaging materials (BN/PPS composites) with high thermal conductivity, excellent dielectric properties, processability, and toughness by conventional melt blending. Si-DPF is located at the interface between the BN fillers and the PPS matrix, which not only improves the dispersion of BN fillers but also strengthens the interaction. With the help of 5 wt% Si-DF, **BN/PPS/Si-DF (70/25/5)** still exhibits the high thermally conductive coefficient (3.985 W/m·K) and low dielectric constant (3.76 at 100 MHz) although BN fillers are loaded as high as 70 wt%. Moreover, the sample processes a lower stable torque value (2.5 N·m), and the area under the stress–strain curves is also increased. This work provides an efficient way to develop high-performance polymer-based composites with high thermally conductive coefficients and low dielectric constants for electronic packaging applications.

## 1. Introduction

Polyphenylene sulfide (PPS) has been widely used as electronic packaging material in the field of electronic communications due to its outstanding thermal dimensional stability, superior chemical resistance, and high flame retardance. With continuous miniaturization and increased power in electronic devices, the comprehensive performance of electronic packaging materials makes more stringent requirements [1,2,3,4]. An ideal electronic packaging material should demonstrate high thermal conductivity, excellent dielectric properties, processability, and toughness. Unfortunately, the PPS matrix has a relatively low thermally conductive coefficient (<0.4 W/m·K), which needs to be enhanced by incorporating thermally conductive particles [5,6,7,8]. Some studies have shown that introducing a large number of thermally conductive particles leads to a dramatic increase in the thermal conductivity of the PPS-based composites, such as boron nitride (BN), expanded graphite (EG), aluminum nitride (AlN), and carbon nanotubes (CNT) [9,10,11,12].

However, a high loading of particles easily forms a large number of agglomerates, deteriorating the processability and toughness of the composites [13,14]. Moreover, the serious agglomeration of particles also leads to more interfacial thermal barriers, which is not conducive to the fabrication of composites with high thermally conductive coefficients and low dielectric constants [15]. Thus, a major challenge in the fabrication of ideal electronic packaging materials (PPS-based composites) is to achieve the uniform dispersion of thermally conductive particles in the PPS matrix.

Usually, the PPS-based composites with high thermal conductivity are prepared by a hot-compression technique in which the surface-modified particles are premixed with the PPS matrix, which greatly improves the dispersion of the particles [16,17,18,19]. Yang et al. [16] modified BN fillers using a silanization reaction with γ-aminopropyl triethoxy silane/aminopropyllsobutyl polyhedraloligomeric silsesquioxane (f-BN) and found that f-BN fillers had better dispersion in the PPS matrix. The thermally conductive coefficient (λ) value of the f-BN/PPS composite with 60 wt% f-BN was improved to 1.122 W/m·K, four times higher than that of the PPS matrix (0.286 W/m·K). Ryu et al. [19] reported that the grafting of (3-Aminopropyl) triethoxysilane onto the surface of BN fillers reduced agglomeration. The λ value of 3.09 W/m·K was exhibited by the modified BN containing 60 wt%. However, due to the complexity of this method, it cannot be extended to large-scale industrial production. Therefore, the uniformly dispersed particles in the PPS matrix could be easily realized by melt blending technology, which is of prime importance for the development of ideal electronic packaging materials in industry.

In this work, a fluorine-containing flow modifier (Si-DF) is designed and synthesized (Figure 1), which is introduced into BN/PPS composites (the weight ratio of BN to PPS is 60:40 and 70:30) by a conventional melt processing technique. The -CF_3_ terminal groups endow Si-DF with low surface energy, which enables Si-DF to locate at a two-phase interface [14]. It is expected that the BN fillers will be dispersed in the PPS matrix, and the BN/PPS composites will present high thermal conductivity, excellent processability, and toughness. The composite will also maintain a low dielectric constant with the help of Si-DF. In addition, the dispersion mechanism is explored.

## 2. Experimental

### 2.1. Main Materials

Polyphenylene sulfide (PPS) was purchased from Nanjing Deyuan Science and Technology Co. Ltd. (Nanjing, China) Boron nitride (BN, mean diameter of 20–30 μm, thermally conductive coefficient of 33 W/m·K) was received from Qinhuangdao ENO High-Tech Materials Development Co., Ltd. (Qinhuangdao, China) Amino silicone oil was purchased from Shin-Etsu Chemical Industrial Co. Ltd. (Tokyo, Japan) 9,10-dihydro-9-oxa-10-phosphaphenanthrene 10-oxide (DOPO) and 3-(Trifluoromethyl) benzaldehyde were purchased from Shanghai Aladdin Biochemical Co. Ltd. (Shanghai, China).

### 2.2. Synthesis of Si-DF

Amino silicone oil (17.2 g, 0.02 mol), 3-(Trifluoromethyl) benzaldehyde (6.96 g, 0.04 mol), and 100 mL of ethanol were added sequentially to a 250 mL round-bottom glass flask equipped with a nitrogen inlet, a condensing device, and a magnetic stirrer. The mixture was stirred at 80 °C for 8 h until the complete conversion of amino silicone oil and 3-(Trifluoromethyl) benzaldehyde, as confirmed by FTIR, and then the intermediate (Si-CF) was obtained. DOPO (8.64 g, 0.04 mol) was added into the system and then further reacted for 12 h at room temperature. After the complete conversion of DOPO was tested via FTIR, the ethanol was removed using a rotary evaporator, and the residual solid was recrystallized from ethyl acetate to yield a yellowish powder named Si-DF (30.76 g, yield: 93.8 %). GPC data: *M*_n_ = 1398 g/mol, PDI = 1.22.

### 2.3. Preparation of BN/PPS Composites

BN and PPS were dried in a vacuum oven at 90 °C for 12 h. The BN/PPS composites were prepared by melt blending in a twin-screw extruder, and the temperatures of the extruder were set in the range from 265 °C to 295 °C at a screw speed of 120 rpm. After being granulated and dried, the pellets were injection-molded on an injection-molding machine to obtain different specimens. The molding was conducted at zone temperature profiles of 275-280-290-290-295 °C. The composite preparation process and composition are simply illustrated in Figure 1, and the compositions and code names of the samples are listed in Table 1.

### 2.4. Characterization

The FTIR spectra (scanned between 400 and 4000 cm^−1^) were obtained on a Nicolet 50XC spectrometer (Nicolet, Glendale, WI, USA). The ^1^H NMR and ^31^P NMR spectra were recorded on an AV400 unity spectrometer (Bruker, Billerica, MA, USA) operated at 400 MHz with DMSO-*d*_6_ as a solvent. The number-averaged molar mass (*M*_n_) and polydispersity (*M*_w_/*M*_n_) of Si-DF were measured by gel permeation chromatography (Waters, Milford, MA, USA). The contact angles were determined on a DSA100 contact angle instrument (KRUSS, Hamburg, Germany) with water and diiodomethane at 25 °C, respectively. Scanning electron microscopy (SEM) images were obtained on a Nova NanoSEM430 scanning electron microscope (FEI, Hillsboro, OR, USA). The torque values of the samples were measured on a Haake torque XSS-300 rheometer (Haake, Vreden, Germany) at a rotation of 50 rpm at 285 °C for 10 min. The thermally conductive coefficient values of the samples were measured on a TPS2200 Hot Disk instrument (AB Corporation, Västerås, Sweden). Dielectric constant values of the samples were measured on a high-frequency Q instrument of QBG-3D (Shanghai Aiyi Electronic Equipment, Shanghai, China). According to ISO 527-2 standard, the tensile strengths of the samples were tested on an AGS-10KNI universal material testing machine (Shimadzu, Kyoto, Japan) at the speed of 10 mm/min.

## 3. Results and Discussion

### 3.1. Characterization of Si-DF

During a typical synthetic process, the intermediate Si-CF is first synthesized via a Schiff base reaction between amino silicone oil and 3-(Trifluoromethyl) benzaldehyde in an EtOH solution. The flow modifier Si-DF is then performed through the addition reaction of DOPO and Si-CF. Figure 2 shows the infrared spectra of Si-CF and Si-DF. The synthesis of the Si-CF intermediate is performed by the condensation of amino silicone oil and 3-(Trifluoromethyl) benzaldehyde. Obviously, the strong peak of -CH=N- (Schiff base) appears at 1650 cm^−1^, and the peak of -CF_3_ appears at 1650 cm^−1^, which indicates the triggered aldimine polymerization between amino silicone oil and 3-(Trifluoromethyl) benzaldehyde. The Si-DF is performed through the addition reaction of DOPO and Si-CF. Additionally, the peak of -CH=N- is absent at the same wavenumber for Si-DF, indicating that DOPO has successfully carried out the addition reaction with the intermediate containing the -CH=N- group [14,20,21,22].

The ^1^H NMR spectrum of Si-DF is displayed in Figure 3a. The peaks at −0.12–0.16 ppm, 0.32–0.46 ppm, and 1.16–1.42 ppm are attributed to the protons in -Si-CH_3_, Si-CH_2_-, and -CH_2_-CH_2_-, respectively. The peaks at 5.29–5.32 ppm and 5.07–5.11 ppm are attributed to the protons in -NH- and P-CH. The peaks at 6.5–8.2 ppm are attributed to the protons in the benzene ring. The ^31^P NMR spectra of DOPO and Si-DF are shown in Figure 3b. Based on the ^31^P NMR spectrum of DOPO, the peaks at 14.40–15.75 ppm are observed, while the peaks are absent at the same wavenumber for Si-DF, and new peaks are observed at 31.50–34.85 ppm. The results from FTIR, ^1^H NMR, and ^31^P NMR show that the flow modifier Si-DF has been successfully synthesized.

### 3.2. Phase Morphology

For highly filled polymer-based composites, how to disperse the fillers in the matrix is a key issue presenting a challenge in the fabrication of high-performance composites. To display the effect of Si-DF on the dispersion of BN fillers, the morphology of the BN fillers in the samples is observed by SEM (Figure 4).

There exists serious agglomeration of BN fillers in the sample **BN/PPS (60/40)** (Figure 4a_1_) and **BN/PPS (70/30)** (Figure 4b_1_). With the addition of Si-DF, the morphology of the BN fillers presents a striking contrast. It is obviously found that BN fillers exhibit a homogeneous dispersion in the samples **BN/PPS/Si-DF (60/35/5)** (Figure 4a_2_) and **BN/PPS/Si-DF (70/25/5)** (Figure 4b_2_).

In our previous work, we demonstrated that the fluorine-containing flow modifiers (PMFs) with low surface energy were located at the MH/LLDPE interface in the highly filled MH/LLDPE composites (80:20 by weight), which improved the dispersion of MH particles [23]. Meanwhile, the distribution of flow modifiers can be estimated by evaluating the wetting coefficient (*ω_a_*) [24]. Herein, Young’s model is used to calculate the *ω_a_* of Si-DF. Additionally, when −1 < *ω_a_* < 1, Si-DF is preferred to locate at the interface. The surface energy of the individual components (PPS, BN, and Si-DF) is examined by measuring the contact angles, and the interfacial energies are calculated using the surface energy of the individual components using the geometric-mean equation or harmonic-mean equation [25,26,27]. All of the data are listed in Table 2 and Table 3, respectively.

As seen, the values of *ω*_a_ for Si-DF are −0.37 and −0.31, respectively, as calculated from the geometric-mean equation, the harmonic-mean equation, and Young’s model, indicating that Si-DF tends to locate at the interface between the BN fillers and PPS matrix. Combining with the phase morphology and theoretical calculations, it is concluded that the distribution of Si-DF at the two-phase interface is effectively able to reduce the formation of agglomeration in the matrix.

### 3.3. Thermal Conductivity

Figure 5 presents the thermally conductive coefficient (λ) values of the PPS matrix and BN/PPS composites. The λ value of the PPS matrix is 0.372 W/m·K. The composites reveal a dramatic enhancement in thermal conductivity in comparison with the PPS matrix. The λ values of **BN/PPS (60/40)** and **BN/PPS (70/30)** are 3.379 W/m·K and 3.569 W/m·K, which are approximately 9 and 9.6 times higher than that of the PPS matrix, respectively. With the addition of Si-DF, the samples present higher thermal conductivity, and the corresponding λ values are increased to 3.873 W/m·K of **BN/PPS/Si-DF (60/35/5)** and 3.985 W/m·K of **BN/PPS/Si-DF (70/25/5)**, respectively. This implies that efficient thermal transfer pathways could be formed at a high loading of BN fillers in the PPS matrix by the introduction of Si-DF [16,28]. Additionally, it is believed that the homogeneous dispersion of BN fillers in the PPS matrix is the key to the formation of thermal transfer pathways. Furthermore, Si-DF forms a protective layer on the BN surfaces, which shows relatively lower interfacial thermal barriers with the PPS matrix compared to the samples without Si-DF, resulting in higher λ values.

Table 4 summarizes the reported thermally conductive coefficient and thermal conductivity enhancement for the BN/PPS composites. It is noted that the BN/PPS composites with higher BN filler contents in this work show a high thermally conductive coefficient and the highest thermal conductivity enhancement. This work provides a relatively more efficient and facile method to improve the thermal conductivity of BN/PPS composites.

### 3.4. Dielectric Property

As a key parameter of electronic packaging materials, a low dielectric constant is conducive to reducing the signal propagation time in electronic components, which is very important in practical applications. The dielectric constant-frequency curves of the PPS matrix and BN/PPS composites are displayed in Figure 6. The PPS matrix possesses a dielectric constant value of 3.32 at 100 MHz. It can be seen that the dielectric constant value of the samples is increased as the loading of BN fillers, and the corresponding values are increased to 3.75 for **BN/PPS (60/40)** and 3.98 for **BN/PPS (70/30)**. It is worth noting that the introduction of Si-DF is beneficial in decreasing the dialectic constant of the samples. For samples **BN/PPS/Si-DF (60/35/5)** and **BN/PPS/Si-DF (70/25/5)**, the dielectric constant values decreased to 3.55 and 3.76, respectively. The lower dielectric constant should be attributed to the weakening of interface polarization [29,30]. Si-DF not only improves the dispersion of BN fillers but also strengthens the interaction between the BN fillers and PPS matrix, weakening the polarization between the BN fillers and PPS matrix, resulting in relatively lower dielectric constant values of the BN/PPS composites.

### 3.5. Processability

The processability of BN/PPS composites is evaluated by the torque rheology test [31]. Figure 7 shows the torque vs. time curves for the BN/PPS composites. As expected, the stable torque values of the samples are increased with the loading of BN fillers, and a pronounced decrease in the torque values of the BN/PPS composites with the addition of Si-DF is observed. For samples **BN/PPS (60/40)** and **BN/PPS (70/30)**, the stable torque values are 3.0 N·m and 11.7 N·m. When 5 wt% of Si-DF is added, the corresponding values are decreased to 2.1 N·m of **BN/PPS/Si-DF (60/35/5)** and 2.5 N·m of **BN/PPS/Si-DF (70/25/5)**, respectively, indicating that Si-DF has an obvious advantage in improving the processability of the BN/PPS composites. Si-DF tends to be located at the interface and effectively reduces the melt viscosity, which prevents BN fillers from aggregation and decreases the friction between the BN fillers and between the BN fillers and the PPS matrix.

### 3.6. Mechanical Properties

Mechanical properties are critical for composites, especially tensile strength and toughness. In this work, tensile testing of the PPS matrix and BN/PPS composites is conducted, and the results are shown in Figure 8. The tensile strength of the PPS matrix is 38 MPa. With the addition of BN fillers, the tensile strength of the samples increased, and the corresponding strength increased to 48 MPa for **BN/PPS (60/40)** and 43 MPa for **BN/PPS (70/30)**. The tensile strength is slightly decreased for the samples with Si-DF. The toughness of the samples is evaluated by calculating the fracture work derived from the area under the stress–strain curve [32,33]. The work of the fracture of the PPS matrix is 13.66 MJ/m^3^, and the work of the fracture is decreased with the loading of BN fillers. For samples **BN/PPS (60/40)** and **BN/PPS (70/30)**, the work of the fracture decreased to 7.46 MJ/m^3^ and 5.53 MJ/m^3^, respectively. This is understandable because BN fillers aggregate easily and form agglomerations in the PPS matrix. As expected, after adding 5 wt% of Si-DF, the work of the fracture of the samples increased to 11.05 MJ/m^3^ of **BN/PPS/Si-DF (60/35/5)** and 8.97 MJ/m^3^ of **BN/PPS/Si-DF (70/25/5)**, indicating that Si-DPF has an obvious advantage in improving the toughness of the highly filled BN/PPS composites. This result is mainly ascribed to the distribution of Si-DF; that is, Si-DPF tends to be located at the interface and forms a “barrier layer” for BN fillers. Si-DF not only reduces the agglomeration but also transfers the external force received by the samples to the small aggregate of BN filler, which is able to act as the stress concentration points to dissipate the external energy, and the toughness of the sample is achieved.

## 4. Conclusions

A fluorine-containing flow modifier Si-DF was successfully synthesized and applied to the fabrication of ideal electronic packaging materials (BN/PPS composites) by conventional melt blending techniques. Si-DF with low surface energy was dominantly located at the BN/PPS interface, leading to the relatively homogeneous dispersion of the BN fillers, thus successfully achieving simultaneously high thermal conductivity, excellent dielectric properties, processability, and toughness in BN/PPS composites, solving the well-known problem that polymer-based composites with high loading thermally conductive fillers usually present high thermal conductivity but poor processability and toughness due to the unavoidable aggregation of the fillers. The Si-DF located at the interface effectively constructed thermal conductive paths and reduced the interfacial heat resistance between the BN fillers and the PPS matrix as well. Meanwhile, the carefully developed additive weakened the polarization between the fillers and the matrix, which reduced the dielectric constant of the composites. In addition, the BN/PPS exhibited excellent processability and toughness. Most importantly, this work paved the way for constructing ideal electronic packaging materials through the rational design of flow modifiers.

## Data Availability

The data that support the findings of this study are available from the authors upon reasonable request.

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
