# Peer review of "Fluorine-Containing Flow Modifier for BN/PPS Composites Enabled by Low Surface Energy"

_molecules, 2022, doi:10.3390/molecules27228066_

Round 1

Reviewer 1 Report

In this manuscript, the BN/PPS composites were fabricated using a fluorine-containing flow modifier (Si-DF). The morphology, thermal, dielectric and mechanical properties of BN/polymer composites were evaluated. This work was well organized, which could be accepted after addressing the following issues.

1.     Is there any adding order for the mixing of BN, PPS and modifier?

2.     The Schiff base reaction of aldehydes (3-(Trifluoromethyl) benzaldehyde) and amines (amino silicone should be further discussed (Ceramics International, 48 (2022) 25833-25839. ACS Materials Letters, (2022) 1787-1797.)

3.     The comparison of the performance between the reported BN based thermal interface materials should be discussed to highlight the superiority of this work.

Reviewer 2 Report

In general, the paper proposed by Authors seems to be valuable and interesting work. Authors should only make some minor revisions and improvements. All recommendations are briefly presented below.

-       Is there any reason to select amount on guest/host ratios in Table 1 and lines 68-69? Please more clarify in this case.

-       It is suggested to input a schematic on how loading BN filler into PPS for in preparation method.

-       The conclusion section can be refined better. It is a simple repetition of the results ... it adds nothing to what has already been said.
